# Chemophotothermal Combined Therapy with 5-Fluorouracil and Branched Gold Nanoshell Hyperthermia Induced a Reduction in Tumor Size in a Xenograft Colon Cancer Model

**DOI:** 10.3390/pharmaceutics17080988

**Published:** 2025-07-30

**Authors:** Sarah Eliuth Ochoa-Hugo, Karla Valdivia-Aviña, Yanet Karina Gutiérrez-Mercado, Alejandro Arturo Canales-Aguirre, Verónica Chaparro-Huerta, Adriana Aguilar-Lemarroy, Luis Felipe Jave-Suárez, Mario Eduardo Cano-González, Antonio Topete, Andrea Molina-Pineda, Rodolfo Hernández-Gutiérrez

**Affiliations:** 1Centro de Investigación y Asistencia en Tecnología y Diseño del Estado de Jalisco, Unidad de Biotecnología Médica y Farmacéutica, Laboratorio de Investigación Traslacional de Terapias Contra el Cáncer, Guadalajara 44270, Mexico; saochoa_al@ciatej.edu.mx (S.E.O.-H.); kavaldivia_al@ciatej.edu.mx (K.V.-A.); 2Laboratorio Biotecnológico de Investigación y Diagnóstico, Departamento de Clínicas, Centro Universitario de los Altos, Universidad de Guadalajara, Tepatitlán 47620, Mexico; yanet.gutierrez@academicos.udg.mx; 3Centro de Investigación y Asistencia en Tecnología y Diseño del Estado de Jalisco, Unidad de Biotecnología Médica y Farmacéutica, Unidad de Ensayos Preclínicos, Guadalajara 44270, Mexico; acanales@ciatej.mx; 4Centro de Investigación Biomédica de Occidente, Instituto Mexicano del Seguro Social, Guadalajara 44330, Mexico; veronicach73@gmail.com (V.C.-H.); adry.aguilar.lemarroy@gmail.com (A.A.-L.); luis.jave@academicos.udg.mx (L.F.J.-S.); 5Centro Universitario de la Ciénega, Universidad de Guadalajara, Ocotlán 47820, Mexico; mario.cano@academicos.udg.mx; 6Centro Universitario de Ciencias de la Salud, Universidad de Guadalajara, Guadalajara 44340, Mexico; antonio.topete.camacho@usc.es; 7Grupo de Física de Coloides y Polímeros, Departamento de Física de Partículas, Facultad de Física, Instituto de Materiales (IMATUS), Universidad de Santiago de Compostela, 15782 Santiago de Compostela, Spain

**Keywords:** hyperthermia, gold nanoparticles, xenograft model antitumor assays

## Abstract

**Background/Objectives**: The heterogeneity of cancer disease and the frequent ineffectiveness and resistance observed with currently available treatments highlight the importance of developing new antitumor therapies. The properties of gold nanoparticles, such as their photon-energy heating, are attractive for oncology therapy; this can be effective and localized. The combination of chemotherapy and hyperthermia is promising. Our aim was to evaluate the combination therapy of photon hyperthermia with 5-fluorouracil (5-FU) both in vitro and in vivo. **Methods**: This study evaluated the antitumor efficacy of a combined chemo-photothermal therapy using 5-fluorouracil (5-FU) and branched gold nanoshells (BGNSs) in a colorectal cancer model. BGNSs were synthesized via a seed-mediated method and characterized by electron microscopy and UV–vis spectroscopy, revealing an average diameter of 126.3 nm and a plasmon resonance peak at 800 nm, suitable for near-infrared (NIR) photothermal applications. In vitro assays using SW620-GFP colon cancer cells demonstrated a ≥90% reduction in cell viability after 24 h of combined treatment with 5-FU and BGNS under NIR irradiation. In vivo, xenograft-bearing nude mice received weekly intratumoral administrations of the combined therapy for four weeks. The group treated with 5-FU + BGNS + NIR exhibited a final tumor volume of 0.4 mm^3^ on day 28, compared to 1010 mm^3^ in the control group, corresponding to a tumor growth inhibition (TGI) of 100.74% (*p* < 0.001), which indicates not only complete inhibition of tumor growth but also regression below the initial tumor volume. Thermographic imaging confirmed that localized hyperthermia reached 45 ± 0.5 °C at the tumor site. **Results**: These findings suggest that the combination of 5-FU and BGNS-mediated hyperthermia may offer a promising strategy for enhancing therapeutic outcomes in patients with colorectal cancer while potentially minimizing systemic toxicity. **Conclusions**: This study highlights the potential of integrating nanotechnology with conventional chemotherapy for more effective and targeted cancer treatment.

## 1. Introduction

Cancer remains one of the leading causes of mortality and morbidity worldwide. [1]. The heterogeneity of the disease and the frequently observed resistance to currently available treatments highlight the importance of developing new therapies with enhanced efficacy and fewer side effects [2,3]. In this context, the application of nanomaterials for the treatment and diagnosis of cancer has emerged as a promising alternative. [4]. The biomedical use of nanomaterials relies on their capacity to function as carriers of bioactive agents, as well as on their ability to act as therapeutic and diagnostic devices.

Gold nanoparticles, particularly gold nanoshells, have been extensively studied as theragnostic elements because of their remarkable optical properties and biocompatibility [5]. Upon irradiation with light, gold nanoshells can convert incident photonic energy into heat, increasing the local temperature of the surrounding medium and inducing damage to malignant tumors. This effect, known as plasmonic photothermal therapy (PTT), has been employed in preclinical studies to achieve highly localized therapy in a spatiotemporally controlled manner [6].

Gold nanoshells are anisotropic structures consisting of a dielectric core surrounded by a thin layer of gold. The precise control of their external radius and shell thickness allows fine-tuning of their localized surface plasmon resonance (LSPR), thereby maximizing light-to-heat conversion at specific wavelengths [7]. Moreover, positioning the plasmon peak wavelength within the near-infrared (NIR) region of the electromagnetic spectrum is particularly desirable, as biological tissues exhibit their lowest extinction coefficients in this spectral range, enabling deeper light penetration [8].

On the other hand, marked improvements have been observed under combined treatment with chemotherapy and hyperthermia in both preclinical and clinical studies. This approach is based on the ability of elevated temperatures to increase drug penetration, increase cellular uptake, and potentiate the cytotoxic effects of chemotherapeutic agents [9]. Hyperthermia has been shown to induce changes in the tumor vasculature, improving drug delivery while also promoting apoptosis and inhibiting DNA repair mechanisms, thereby sensitizing cancer cells to chemotherapy [10]. Specifically, the combination of 5-fluorouracil (5-FU) and hyperthermic therapy has been evaluated in gastric cancer cells and demonstrated improved induction of apoptosis [11]. In addition, hyperthermic intraperitoneal chemotherapy with a combination of mitomycin, 5-FU, and oxaliplatin markedly inhibited the growth of colorectal cancer cells under hyperthermic conditions in vitro and appeared to be safe and feasible for patients at high risk of colorectal peritoneal metastasis [12]. However, the simultaneous application of chemotherapeutic drugs with plasmonic photothermal therapy (PTT) remains largely unexplored. Given the precise and localized heating capabilities of gold nanoshells, their integration with chemotherapeutic agents could provide a promising strategy to enhance treatment efficacy while minimizing systemic toxicity.

In this work, we synthesized branched gold nanoshells (BGNSs) via a seed-mediated growth method. The nanoshells exhibited an LSPR peak at 800 nm, rendering them highly photoresponsive to laser stimulation and endowing them with optimal hyperthermic therapeutic properties. The BGNSs were characterized, and their cytotoxicity against a fluorescent colon cancer cell line was determined. Furthermore, the antitumor effect of BGNSs, when intratumorally coadministered with 5-FU, was evaluated in a xenograft colon cancer murine model. The results revealed enhanced tumor growth inhibition in the combined chemotherapy group compared to the individual treatment groups. These findings suggest that combination therapy may be a promising strategy to improve treatment outcomes for this type of cancer.

## 2. Materials and Methods

### 2.1. Synthesis of BGNSs

The BGNS synthesis methodology is based on the methods of Topete et al. (2014) [13]. The first step involves preparing a mixture of chitosan-modified PLGA nanoparticles and gold seeds, resulting in PLGA-gold seed precursors. The second step consists of the formation of the shell and its pegylation. PLGA-gold seed precursors are mixed with a Au + 1 growth solution. Subsequently, these precursors are combined with PEG, resulting in BGNSs.

### 2.2. Physical and Chemical Characterization of the BGNSs

The hydrodynamic size and zeta potential of the BGNSs were characterized using a Zetasizer ZS90 analyzer from Malvern Panalytical (Malvern, Worcestershire, UK). For this, 100 µL of BGNSs was placed in polystyrene cells and measured at a 90° angle for ten series of 60 s in triplicate. BGNS size and morphology were analyzed by scanning transmission electron microscopy (STEM) with an FE-SEM JSM-7800F (JEOL, Tokyo, Japan) microscope equipped with a STEM detector at 30 kV (Deben, UK Ltd., Bury St. Edmunds, UK) and via transmission electron microscopy (TEM) with a Phillips CM-12 microscope at 120 kV. Optic extinction was measured with a UV–vis V-730 (Jasco, Hachioji, Japan) spectrophotometer.

### 2.3. Cell Culture

The cell line used in this work was the SW620-GFP colon cancer cell line (AntiCancer Inc., San Diego, CA, USA). It was first isolated from the large intestine of a 51-year-old white male patient with Dukes C (stage III) colorectal cancer. The cells were maintained in RPMI 1640 medium supplemented with 10% fetal bovine serum (FBS) and 1% penicillin–streptomycin. The culture conditions were 37 °C in an atmosphere of 5% CO_2_ and 95% relative humidity (RH).

### 2.4. In Vitro Cytotoxicity Assays of Chemotherapy

5-FU, with the commercial name “Carebin,” was used at a concentration of 250 mg/10 mL in solution, from the pharmaceutical company PiSA (Batch L175039; Guadalajara, Mexico), and stored at room temperature in the dark. A 1 mM stock solution of 5-FU was prepared in RPMI-1640 medium, which was then used to create different concentrations.

To determine the cytotoxic effect of 5-FU, an MTT cell viability assay was performed in 96-well plates. A total of 2 × 10^4^ cells were seeded well and incubated overnight. After monolayer formation, the cells were treated with different concentrations of 5-FU: 12.5, 25, 50, 100, 150, 200, 250, 300, 350, and 400 μM. Cell viability was measured after 24 h of exposure to the chemotherapeutic agent. Next, the RPMI-1640 medium was removed, and the cells were washed with PBS. Then, 100 μL of RPMI-1640 and 10 μL of 5 mg/mL MTT solution (Sigma-Aldrich, St. Louis, MO, USA) were added, and the cells were incubated for 3 h at 37 °C. After the MTT solution was removed, the remaining crystals in the wells were solubilized. The absorbance was determined at a wavelength of 570 nm. To determine the concentration of 5-FU necessary to achieve 50% cytotoxic activity, the half-maximal inhibitory concentration (IC_50_) was measured. Each experiment was repeated at least three times in triplicate.

### 2.5. In Vitro Assays of Hyperthermia

To evaluate hyperthermia, it was necessary to verify that the BGNS increased the temperature when irradiated. For this purpose, an initial 1:1 dilution was prepared by mixing 100 μL of BGNS with 100 μL of RPMI-1640 medium. A control of 200 μL of deionized water was also prepared in 48-well plates. These solutions were irradiated for five minutes at 2 watts (W) and 800 nm, after which the temperature was subsequently measured. The use of deionized water as a control was part of the thermal standardization phase, allowing us to confirm that the baseline temperature started at approximately 36 ± 0.5 °C without interference from medium components. This ensured accurate monitoring of the temperature increase induced by BGNS under NIR irradiation. Subsequent experiments used BGNS diluted in RPMI-1640 medium to evaluate the biological effects under physiologically relevant conditions.

After verifying that the BGNSs fulfilled their function, different dilutions and incubation times were tested. For this purpose, 8 × 10^4^ cells per well were seeded in 48-well plates, which were then incubated for 24 h. After the formation of the monolayer, different dilutions of BGNSs were added: 1:5—(1.46 × 10^15^ NPs/mL), 1:10—(7.30 × 10^14^ NPs/mL), and 1:20—(3.65 × 10^14^ NPs/mL). The cells were subsequently incubated for (a) 24 h and (b) 3 h, before being irradiated at 2 W and 800 nm for six minutes. Finally, the MTT assay was performed to evaluate cell viability.

After the optimal incubation time was determined, the amount of BGNS used in the treatment was standardized. In the previous experiment, dilutions of 1:5, 1:10, and 1:20 were used, and it was observed that dilutions higher than 1:10 might not yield the expected results. Therefore, in this experiment, only dilutions between 1:5 (1.46 × 10^15^ NPs/mL) and 1:10 (7.30 × 10^14^ NPs/mL) were used. Once again, 8 × 10^4^ cells per well were placed in 48-well plates and allowed to grow until a monolayer formed, after which the corresponding dilutions were added. The cells were incubated for an additional 3 h, followed by irradiation at 2 W and 800 nm for six minutes before the MTT assay was performed. For both experiments, the initial and final temperatures (before and after irradiation, respectively) were measured to verify that the initial and final temperatures reached 43 ± 0.5 °C.

### 2.6. Chemophotothermal In Vitro Assays

In a 48-well plate, 8 × 10^5^ cells were seeded and incubated until a monolayer formed. Negative and positive controls were included, along with individual controls for 5-FU, BGNS, and irradiation. Additionally, controls for irradiated BGNS and the entire treatment were included, which consisted of a 1:10 dilution (7.30 × 10^14^ NPs/mL) of BGNS incubated for 3 h, followed by irradiation at 2 W, 800 nm for 10 min, and 75.94 μM 5-FU. The treatment mixture was incubated for 24 and 48 h before the MTT assay was performed.

### 2.7. Chemophotothermal in Vivo Assays

Immunodeficient nude murine models of the Nu/Nu strain, 8-week-old females, were obtained from Bioterio Morelos (Cuauhtempan, Morelos, Mexico) and kept at the Research Vivarium at the Center for Research and Assistance in Technology and Design of the State of Jalisco (CIATEJ). The experiments were conducted according to the guidelines set by the Guide for the Care and Use of Laboratory Animals and the rules established by the CICUAL of CIATEJ, under the approved project number (2023-008A).

The SW620-GFP cells inoculated into the murine models were cultured in RPMI-1640 medium. Upon reaching approximately 85–90% confluence, trypsinization was performed (a technique commonly used to dissociate adherent cells from culture flasks). After dissociation, the cells were mixed with Matrigel (Sigma-Aldrich) at a 50:50 ratio, resulting in a total volume of approximately 120 μL in a microtube. This mixture was collected via a 1 mL insulin syringe with a 29 gauge (G) needle that was 13 mm long. Once the inoculum was prepared, the model mice were anesthetized, and the cells were administered subcutaneously once on each flank. The experiment began by inoculating 5 × 10^6^ cells per inoculation, and the appearance and development of the tumors were monitored daily. After one week, the tumors had grown sufficiently to begin treatment (approximately 2 mm^3^). The tumor growth was not uniform.

Mice were inoculated with tumors and monitored for tumor growth. On day 7, animals presenting with comparable tumor sizes were paired and evenly distributed into five treatment groups (n = 10, with four tumors per group): control, 5-FU, NIR, BGNS + NIR, and 5-FU + BGNS + NIR. Treatments were administered once a week for more than four weeks. The conditions established in the in vitro assays were applied to the in vivo NIR groups at 2 W, 800 nm for ten minutes. According to the literature, administering 10–125 mg/kg of 5-FU is considered the optimal dose for treatment. Using this as a reference, calculations were made based on the concentration of 5-FU used (250 mg/10 mL) and the mean weight of the murine models (22 g), yielding an approximate result of 107 μL (2.6 mg) of the chemotherapeutic agent mixed with PBS.

After the treatments were applied, the murine models were monitored via a small animal imaging system, the UVP iBox Explorer2 from Analytik-Jena, which allows noninvasive detection of fluorescent and bioluminescent indicators, enabling tumor development to be observed. Tumors were measured with a caliper, while the murine models were weighed with a scale before being examined via UVP iBox equipment. Observations with the UVP iBox were performed every four days. Before being placed in the equipment, the models were measured, weighed, and received the corresponding treatment. The tumor volume was calculated via the following equation: V = 0.5 × L × W^2^ [14], where V is the tumor volume, L is the length of the tumor, and W is the weight of the tumor. Before the tumors exceeded 1.5 cm, the model mice were sacrificed, and the tumors were extracted [15].

## 3. Results

### 3.1. Characterization of BGNSs

PLGA NPs, used as cores for the subsequent growth of BGNSs, were synthesized using a nanoprecipitation method. The resulting NPs had a hydrodynamic diameter of 126.3 ± 7.0 nm and a Z-potential of −12.1 ± 1.7 mV, which was provided by the carboxylic groups of the biodegradable copolymer’s glycolic and lactic acid repeating units. Subsequently, the surface charge of the PLGA NPs was shifted towards a positive value (+18.5 ± 1.2 mV) after the adsorption of cationic oligochitosan Table 1. The coating with the oligosaccharide also increased in hydrodynamic size. This surface charge inversion enabled the electrostatic adsorption of 5 nm citrate-stabilized gold seeds, which served as nucleation sites for the subsequent growth of the anisotropic gold layer. To achieve this, PLGA-seed precursors were mixed with a K-gold solution and ascorbic acid, which acted as a reducing agent. The sequence of synthetic steps is schematized in Figure 1a. Finally, BGNS were surface-coated with PEG to increase the colloidal stability of the gold nanostructures. The analysis of the size of PEGylated BGNS revealed an average hydrodynamic diameter of 194.5 nm, as shown in Table 1 and Figure 1b.

The optical extinction of the BGNSs, obtained using a UV–vis spectrophotometer V-730, showed a pronounced LSPR peak centered at ~800 nm, indicating strong absorption in the NIR region, suitable for in vivo photothermal applications (Figure 1c).

The morphology of bare PLGA NPs was spherical, as observed in TEM images (Figure 2a). The surface coating of the polymeric cores with polycationic oligochitosan and the adsorption of the negatively charged citrate-stabilized gold seeds were visualized by TEM (Figure 2b). Morphological analysis of the BGNS by STEM revealed an anisotropic topography with distinctive branches and protrusions (Figure 2c).

The optical extinction results of the BGNSs obtained via a V-730 UV–vis spectrophotometer showed a maximum surface plasmon resonance centered at approximately 800 nm (Figure 2).

### 3.2. In Vitro Cytotoxicity of Chemotherapy and Hyperthermia

The cytotoxicity assay using 5-FU in SW620-GFP colon cancer cells demonstrated a dose- and time-dependent reduction in cell viability. At 30 µM, cell viability was reduced to 72.28% after 24 h and 40.36% after 48 h. At 50 µM, the viability decreased to 64.36% (24 h) and 32.20% (48 h), whereas at 70 µM, the reduction was more pronounced, reaching 46.81% at 24 h and 26.80% at 48 h (Figure 3). These results confirm the cytotoxic potential of 5-FU and support the selection of 70 µM as an effective concentration for subsequent combination therapy assays.

To evaluate the photothermal effect of BGNS, SW620-GFP colon cancer cells were incubated with different BGNS dilutions (1:5, 1:10, and 1:20) and subsequently irradiated with near-infrared (NIR) light at 800 nm and 2 W for six minutes. Cell viability was assessed via the MTT assay. The results revealed an inverse relationship between the BGNS concentration and cell viability: the 1:5 dilution resulted in the highest cytotoxicity (11.88% viability), followed by the 1:10 (43.37%) and 1:20 (67.93%) dilutions (Figure 4a,b). Based on these findings, the 1:10 dilution (corresponding to 3.33 µg AuNPs/mL) was selected as the optimal concentration for subsequent in vitro combination therapy experiments.

To evaluate the efficacy of combined chemo-photothermal therapy, SW620-GFP colon cancer cells were exposed to various treatment conditions, including negative control (untreated cells), positive control (0.1 N hydrochloric acid in isopropanol), 5-FU alone at 70 µM, near-infrared (NIR) irradiation alone, BGNSs with NIR, and the whole combination of 5-FU + BGNS + NIR. The MTT assay revealed that the negative control maintained high cell viability, while the positive control induced complete cytotoxicity. Treatment with 5-FU alone reduced viability to 36.94% after 48 h. NIR irradiation alone had minimal effect, whereas BGNS + NIR reduced viability to 58.21%. Notably, the combined treatment of 5-FU + BGNS + NIR resulted in a dramatic reduction in cell viability to 3.57%, indicating a marked decrease in viability under the combined treatment, as shown in Figure 5.

### 3.3. In Vivo Chemophotothermal Treatment

#### 3.3.1. Standardization

To determine the optimal conditions for in vivo chemo-photothermal therapy, preliminary standardization experiments were conducted using SW620-GFP xenograft-bearing mice. Initial tests began with the administration of 30 µL of 5-FU; however, no observable changes in tumor size were detected after four days. Consequently, the dose was increased fivefold and subsequently tenfold, yet the tumor response remained minimal. These findings underscore the importance of optimizing both the drug concentration and the nanoparticle dosage.

In parallel, BGNSs were tested at various concentrations to evaluate their photothermal efficacy in vivo. The first trial used 15 µL of BGNS, which was progressively increased by factors of 5, 10, 50, and 100. These adjustments were made to determine the minimum effective dose capable of inducing a measurable hyperthermic response under NIR irradiation. The results confirmed that insufficient nanoparticle concentration failed to generate the desired thermal effect, reinforcing the importance of dose optimization.

To assess the spatial and therapeutic effects of different treatment combinations, each mouse was inoculated with three or four tumors in distinct anatomical locations. The treatments were as follows: (1) 5-FU alone, (2) BGNS + NIR, (3) 5-FU + BGNS + NIR, and (4) the untreated control. For the third mouse, the position of each treatment was rotated to minimize anatomical bias. The SW620-GFP cell line was selected for its fluorescent properties, which enabled noninvasive tumor monitoring via the UVP iBox imaging system. To determine tumor sizes, a volumetric measurement using a micrometric caliper (with 10 μm resolution) was used to determine the length (L) and width (W) of the tumor utilizing the formula V = 0.5 × L × W^2^ to estimate the tumor volume, as previously validated in the literature [16].

Each treatment was administered intratumorally, beginning with BGNS, followed by 5-FU, and concluded with NIR irradiation (800 nm, 2 W, 10 min). This sequence was designed to ensure optimal nanoparticle distribution and thermal activation. The standardization phase confirmed that both the concentration and order of administration are critical for achieving reproducible and effective tumor reduction in vivo.

#### 3.3.2. Control Group

The control group was inoculated, and tumor growth was monitored over four weeks. The mice in this group were handled and anesthetized every seven days, and their weight and tumor size were recorded. No therapeutic intervention was applied. Tumor growth was not uniform across individuals. The mice were euthanized before the tumors reached a size of 1.5 cm (Figure 6). The average tumor volume increased progressively: on day 7 (4.8 mm^3^), day 14 (39.33 mm^3^), day 21 (314 mm^3^), and day 28 (1010 mm^3^). Using the day 7 measurement as a reference point, the tumor size on day 28 represented an approximately 210-fold increase in size.

#### 3.3.3. Treatment Group: 5-FU

A dose of 25 mg/kg 5-FU (Carebin, PiSA) was administered intratumorally every 7 days for up to 30 days of follow-up, according to the methodology reported by Sang et al., 2020 [17]. For this purpose, equipment was used to evaporate the liquid from the drug in the vehicle through centrifugation. Upon obtaining the salt in pellet form, it was dissolved in 150 µL of RPMI medium. Compared to the control group, tumor mass growth was constant but not uncontrolled (Figure 7). The average tumor size increased over the following days: day 7 (4.75 mm^3^), day 14 (48 mm^3^), day 21 (78.25 mm^3^), and day 28 (142.5 mm^3^).

The 5-FU group showed a tumor volume increase from 4.75 mm^3^ to 142.5 mm^3^, corresponding to a TGI of 86.30%.

#### 3.3.4. Treatment Group: NIR

Treatment was administered every 7 days up to 30 days of follow-up. The mice were anesthetized and subjected to NIR at an 800 nm wavelength for 10 min at a power of 2 W, with a 1 cm distance between the laser and the tumor (Figure 8). No significant temperature increase was recorded in the irradiated area, as the presence of BGNS is necessary to raise the temperature to 45 ± 0.5 °C. However, the growth was not as accelerated as that in the control group. The average tumor size increased over the following days: day 7 (15.75 mm^3^), day 14 (100.5 mm^3^), day 21 (180.5 mm^3^), and day 28 (472 mm^3^). The NIR-only group increased from 15.75 mm^3^ to 472 mm^3^, corresponding to a TGI of 54.61%.

#### 3.3.5. Treatment Group: NIR + BGNS

Treatment was administered every 7 days up to 30 days of follow-up. The hyperthermia groups were administered 7.30 × 10^14^ BGNS/mL with NIR at 800 nm at 2 W for 10 min, maintaining a 1 cm distance between the laser and the tumor, until the temperature reached 45 ± 0.5 °C (Figure 9). A thermographic camera was used for real-time photographic documentation of the temperature increase. Additionally, the weight of each mouse was monitored via an analytical balance, and the change in tumor size was measured with a Vernier caliper (Figure 10). The average tumor size increased over the following days: day 7 (7.5 mm^3^), day 14 (0.72 mm^3^), day 21 (99.17 mm^3^), and day 28 (303 mm^3^). The BGNS + NIR group grew from 7.5 mm^3^ to 303 mm^3^, corresponding to a TGI of 70.60%.

#### 3.3.6. Treatment Group: 5-FU + NIR + BGNS

Treatment was administered every 7 days up to 30 days of follow-up. This hyperthermia plus chemotherapy group was administered a solution containing 7.30 × 10^14^ BGNS/mL and 25 mg/kg 5-FU (Carebin, PiSA) in 150 µL of RPMI medium intratumorally, with NIR application at 800 nm and 2 W for 10 min at a distance of 1 cm between the laser and the tumor until the temperature increased to 45 ± 0.5 °C (Figure 11). A thermographic camera was used for real-time photographic documentation of the temperature increase. Additionally, the weight of each mouse was monitored via an analytical balance, and the change in tumor size was measured with a Vernier caliper (Figure 12). The average tumor size increased over the following days: day 7 (7.8 mm^3^), day 14 (1.125 mm^3^), day 21 (0.8 mm^3^), and day 28 (0.4 mm^3^). Notably, the combined treatment group (5-FU + BGNS + NIR) demonstrated a reduction in tumor volume from 7.8 mm^3^ to 0.4 mm^3^, resulting in a TGI of 100.74%, indicating tumor regression.

#### 3.3.7. Comparison of the Treatment and Control Groups

The averages of the measurements shown in the previous graphs for each group were compared in a time graph, where the changes in tumor sizes of each group can be observed on days 7, 14, 21, and 28 of the in vivo experimental model. The control group showed uncontrolled tumor growth, reaching a maximum size of 1010 mm^3^; the NIR group had the next most significant tumor size value of 472 mm^3^; the NIR + BGNS group showed a reduction in size after treatment; however, the tumors grew again, reaching a tumor size of 303 mm^3^; the 5-FU chemotherapeutic group had constant but measured growth, maintaining its size at 142.5 mm^3^; the complete treatment group with 5-FU + NIR + BGNS showed a significant reduction in tumor size through the administration of the treatment, with the lowest size value among all the groups of 0.4 mm^3^ (Figure 13 and Figure 14).

To quantitatively assess the therapeutic efficacy of each treatment, the Tumor Growth Inhibition percentage (TGI%) was calculated based on the tumor volume measurements on days 7 and 28. The control group exhibited an increase in tumor volume from 4.8 mm^3^ to 1010 mm^3^. In contrast, the 5-FU group showed a tumor volume increase from 4.75 mm^3^ to 142.5 mm^3^, corresponding to a TGI of 86.30%. The NIR-only group increased from 15.75 mm^3^ to 472 mm^3^ (TGI = 54.61%), while the BGNS + NIR group grew from 7.5 mm^3^ to 303 mm^3^ (TGI = 70.60%). Notably, the combined treatment group (5-FU + BGNS + NIR) demonstrated a reduction in tumor volume from 7.8 mm^3^ to 0.4 mm^3^, resulting in a TGI of 100.74%, indicating tumor regression. The methodology used for TGI% calculation was adapted from Yang et al. (2022), who applied this approach in the similar context of chemo-photothermal therapy for colorectal cancer [15].

## 4. Discussion

The development of novel therapeutic strategies for colon cancer is of paramount importance, and this study explores a promising combination of chemotherapy and hyperthermia mediated by branched gold nanoparticles. The characterization of the BGNSs demonstrated crucial physicochemical properties for their biomedical application. The increase in hydrodynamic diameter and inversion of the zeta potential upon chitosan functionalization are consistent with previous studies, which highlight the ability of this polysaccharide to enhance the colloidal stability and biocompatibility of nanoparticles —essential factors for their in vivo administration and interaction with biological systems [18]. A distinctive aspect of this work lies in the synthesis of the BGNSs, an original technology developed by our research team, stemming from the inventiveness of this group of investigators. The anisotropic morphology observed in the BGNSs could confer additional advantages in cell–nanoparticle interactions, potentially facilitating the internalization and release of therapeutic cargo more efficiently in tumor cells, as suggested by research with other nonspherical nanoparticles. The surface plasmon resonance peak in the near-infrared (NIR) region, centered at approximately 800 nm, is particularly relevant for photothermal therapy, as this spectral window allows for greater light penetration through biological tissues, minimizing absorption by endogenous components such as hemoglobin and water and thereby optimizing localized heat generation in the tumor.

In vitro cytotoxicity assays revealed the effectiveness of cisplatin and 5-FU in reducing the viability of SW620-GFP cells, with a correlation between dose and exposure time, which aligns with their known mechanism of action as chemotherapeutic agents. Notably, the combination of both drugs exhibited an improved efficacy under combined treatment, underscoring the potential of polytherapy strategies to overcome drug resistance and improve treatment efficacy in colon cancer [19]. The ability of the BGNSs to induce cytotoxicity upon irradiation with NIR light, as well as their basal cytotoxic effect independent of light, suggests a dual therapeutic mechanism of action. The localized hyperthermia generated by the BGNSs upon irradiation can irreversibly damage tumor cells, while the presence of the nanoparticles per se could interfere with essential cellular processes. Hyperthermia has been described to induce increased permeability to drugs by affecting membrane fluids, altering the concentrations of ions, such as Ca^2+^, Na^+^, K^+^, and Mg^2+^. Cytoskeletal alterations are likely to occur, which induces increased intracellular drug flow and programmed cell death as well as the production of reactive oxygen species (ROS) in mitochondria. In addition, nanoparticle-induced hyperthermia increases the necrosis and apoptosis of tumor cells [20]. The enhanced therapeutic effect observed with the combination of BGNS-mediated hyperthermia and chemotherapy (specifically 5-FU in the in vivo model) could be attributed to an increase in drug internalization by heat-sensitized cells or the inhibition of cellular repair mechanisms induced by thermal damage.

The results obtained in the in vivo murine model demonstrated the superiority of the combined treatment of 5-FU- and BGNS-mediated hyperthermia with NIR irradiation in reducing subcutaneous tumor growth over individual therapies. This observation is consistent with the growing evidence in the literature supporting the combination of photothermal therapy with chemotherapeutic agents to achieve greater antitumor efficacy [21]. Hyperthermia can increase vascular permeability in tumors, facilitating the delivery of chemotherapeutic drugs to cancer cells, and can also inhibit mechanisms of drug resistance. The selectivity of nanoparticle-mediated hyperthermia, which involves primarily targeting irradiated tumor tissue, has the potential to minimize the systemic side effects associated with traditional chemotherapy. The creation and application of original technologies, particularly in the synthesis of nanoparticles by our team, underscore the capacity for scientific and technological innovation generated by Mexican researchers in the field of oncological nanomedicine.

Although this study did not directly assess the impact of the combined therapy on cancer stem cells (CSCs), it is well established that CSCs play a significant role in tumor recurrence, metastasis, and resistance to conventional treatments. Notably, 5-FU has been reported to enrich CSC populations in colorectal cancer models, potentially limiting its long-term efficacy [22]. However, photothermal therapy has shown promise in sensitizing CSCs by disrupting their protective microenvironment and impairing key survival pathways [23]. Therefore, the localized hyperthermia induced by BGNS-mediated NIR irradiation may enhance the susceptibility of CSCs to 5-FU. Future studies should investigate this hypothesis by incorporating CSC-specific markers and functional assays to assess the impact of this therapeutic strategy on CSC populations.

Although the results are encouraging, it is essential to recognize the limitations of this study. The murine xenograft model, although widely used in preclinical cancer research, does not fully replicate the complexity of the human tumor microenvironment, particularly in terms of immune system interactions and intratumoral heterogeneity. Moreover, the use of nude mice, which lack functional T lymphocytes, introduces an additional limitation, as their immunosuppressed condition precludes the evaluation of immune-related therapeutic responses. This may lead to an underestimation or misrepresentation of the role that the immune system could play in the observed treatment effects. Consequently, the translation of these findings to clinical settings will require validation in more sophisticated preclinical models and, ultimately, in human clinical trials. Future studies should aim to elucidate the molecular mechanisms underlying the enhanced response observed when combining chemotherapy with BGNS-mediated hyperthermia. Additionally, optimizing nanoparticle delivery and irradiation parameters is critical for enhancing therapeutic efficacy while minimizing potential side effects. Investigating the integration of this therapeutic approach with other treatment modalities may also offer a promising avenue for improving outcomes in colon cancer therapy.

## 5. Conclusions

The median lethal doses of the chemotherapeutic agents cisplatin and 5-fluorouracil were determined in the SW620-GFP cell line. Furthermore, the optimal conditions for temperature, exposure time, and concentration of branched gold nanoparticles for hyperthermia treatment were defined in an in vitro model of colon cancer. The successful establishment of an in vivo murine model of subcutaneous colon cancer xenografts enabled the evaluation of the post-treatment therapeutic effect of chemotherapy and hyperthermia, revealing a significant reduction in tumor mass size when tumors were treated with the chemotherapeutic 5-fluorouracil and branched gold nanoparticles irradiated with near-infrared light. This finding suggests the promising combined therapeutic benefit potential, where hyperthermia mediated by branched gold nanoparticles could enhance the efficacy of 5-fluorouracil in reducing the tumor burden in an in vivo model. The determination of the in vitro median lethal doses and the optimization of hyperthermia conditions establish a foundation for future preclinical investigations.

The implications of these results are significant for the field of colon cancer treatment, opening avenues for adjuvant strategies that could improve therapeutic outcomes, with a potential reduction in systemic side effects. While the murine model presents limitations, the findings warrant further research to elucidate the underlying mechanisms, evaluate long-term toxicity, and explore its feasibility as an innovative strategy against this disease. Future studies should focus on more complex models and investigate different administration regimens and therapeutic combinations to advance toward potential clinical translation.

## Figures and Tables

**Figure 1 pharmaceutics-17-00988-f001:**
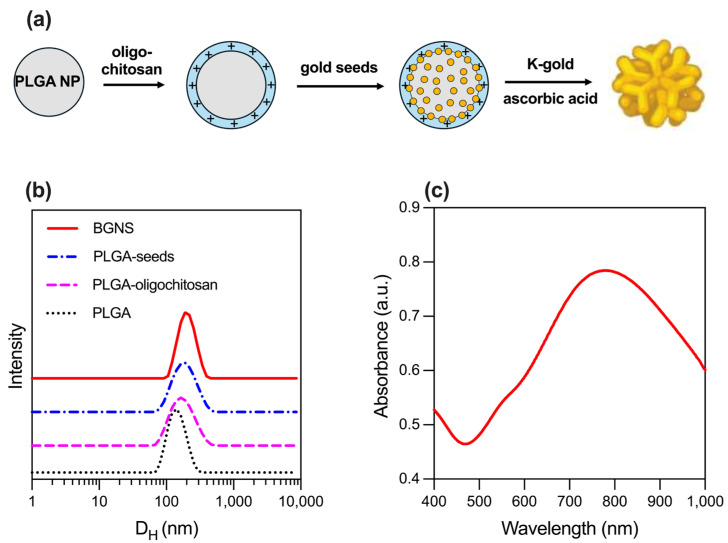
Synthesis and physicochemical characterization of BGNSs. (**a**) Schematics of the seeded-growth synthesis of BGNSs. (**b**) Intensity-averaged size distribution of PLGA, PLGA-oligochitosan, PLGA-seed NPs, and BGNSs. (**c**) UV–vis absorption spectrum of BGNSs with LSPR peak at ~800 nm.

**Figure 2 pharmaceutics-17-00988-f002:**
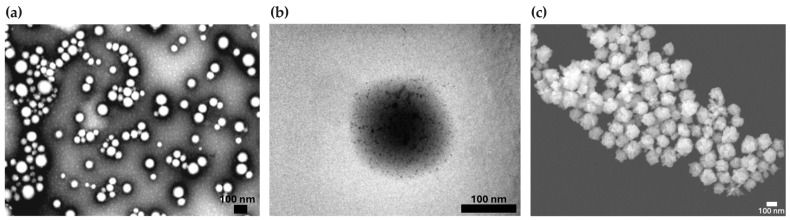
Representative electron microscopy images illustrating the synthesis and morphological characterization of the BGNSs. (**a**) TEM image of PLGA NPs displaying a quasi-spherical morphology. (**b**) TEM image of PLGA-seed precursors, showing the electrostatic adsorption of negatively charged gold seeds onto the positively charged surface of oligochitosan-modified PLGA NPs. (**c**) STEM image of BGNSs, revealing an anisotropic surface topology with distinctive branches and protrusions.

**Figure 3 pharmaceutics-17-00988-f003:**
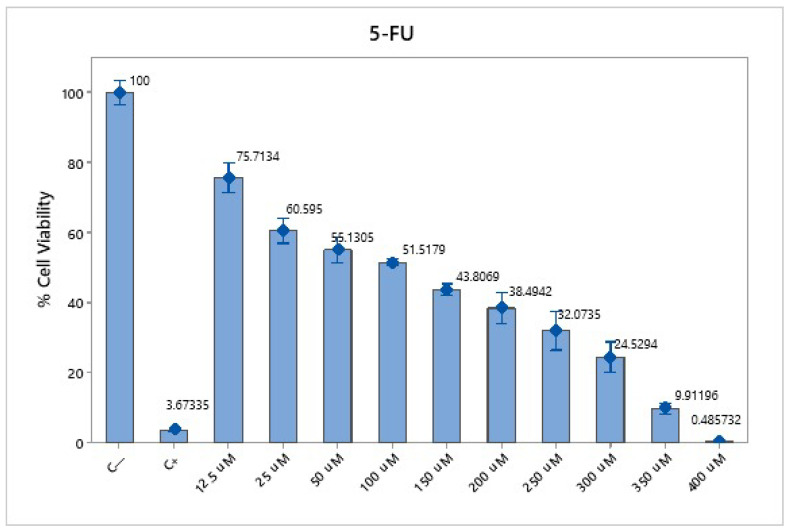
Cytotoxicity of 5-FU in SW620-GFP colon cancer cells. Cell viability percentages after 24 h of exposure to 5-FU at concentrations of 30 µM, 50 µM, and 70 µM. A precise dose- and time-dependent decrease in viability was observed, with the most significant reduction at 70 µM. The negative control consisted of untreated cells, and the positive control was 0.1 N hydrochloric acid in isopropanol. The data represent the means of three independent experiments performed in triplicate.

**Figure 4 pharmaceutics-17-00988-f004:**
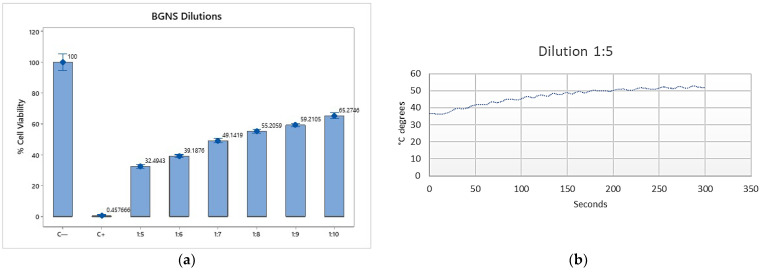
In vitro cytotoxicity of BGNSs under near-infrared (NIR) irradiation. (**a**) SW620-GFP colon cancer cells were treated with BGNSs at dilutions of 1:5, 1:10, and 1:20, followed by NIR laser exposure (800 nm, 2 W, 6 min). Cell viability was assessed using the MTT assay. (**b**) A dose-dependent decrease in viability was observed, with the 1:5 dilution exhibiting the most pronounced cytotoxic effect. Data represent the mean ± standard deviation of three independent experiments performed in triplicate.

**Figure 5 pharmaceutics-17-00988-f005:**
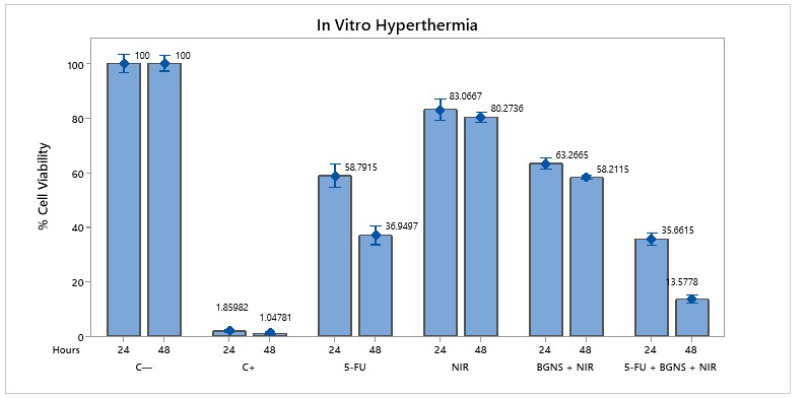
Bar graph comparing the viability of SW620-GFP colon cancer cells under six different treatment conditions after 48 h. The figure illustrates the differential cytotoxic effects of each group, highlighting the minimal impact of NIR alone, the moderate reduction caused by 5-FU and BGNS + NIR, and the pronounced enhanced therapeutic effect of the combined 5-FU + BGNS + NIR treatment. The error bars indicate SD from triplicate experiments.

**Figure 6 pharmaceutics-17-00988-f006:**
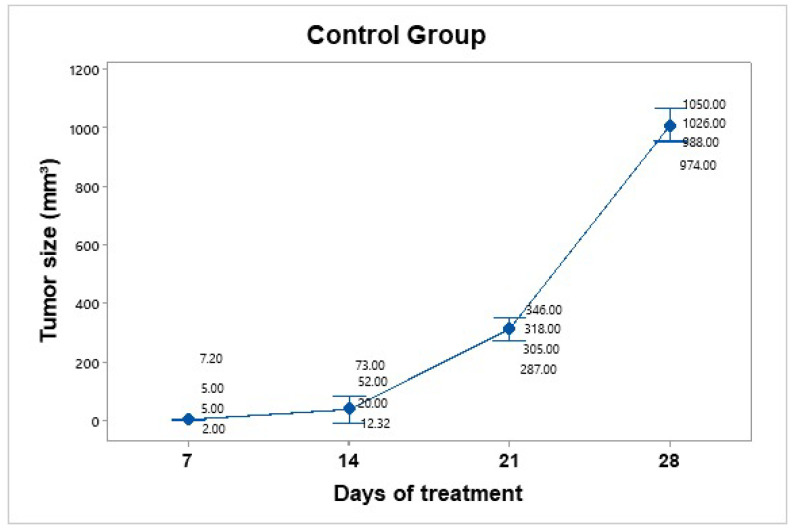
Tumor growth progression in the control group over 28 days. The graph displays individual tumor measurements and the group mean (blue dots) at each time point. A consistent and exponential increase in tumor size was observed, reaching an average of 1010 mm^3^ by day 28. Data dispersion is shown using a 95% confidence interval based on Student’s t distribution. This group served as a baseline for evaluating the efficacy of treatment.

**Figure 7 pharmaceutics-17-00988-f007:**
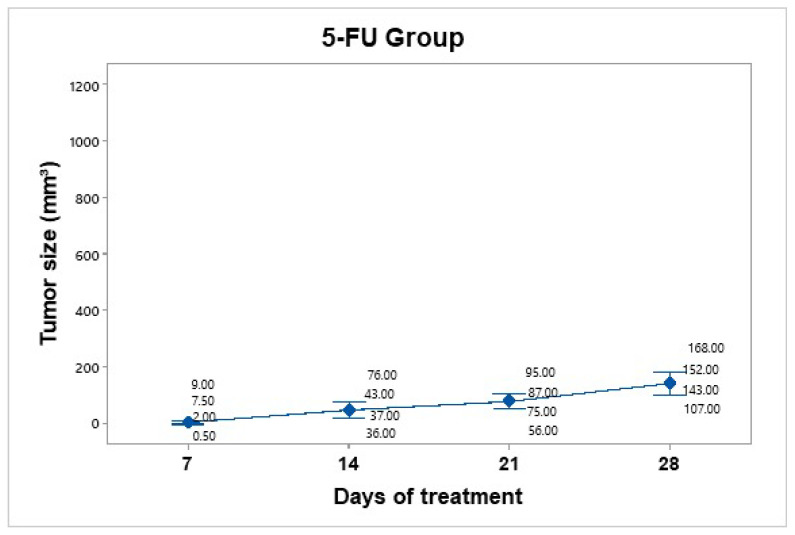
Tumor volume progression in the 5-FU treatment group over 28 days. The graph shows individual tumor measurements and the group mean (blue dots) at each time point. Compared to the control group, tumor growth was slower and more controlled, reaching an average of 142.5 mm^3^ by day 28. Data are presented with a 95% confidence interval via Student’s t distribution.

**Figure 8 pharmaceutics-17-00988-f008:**
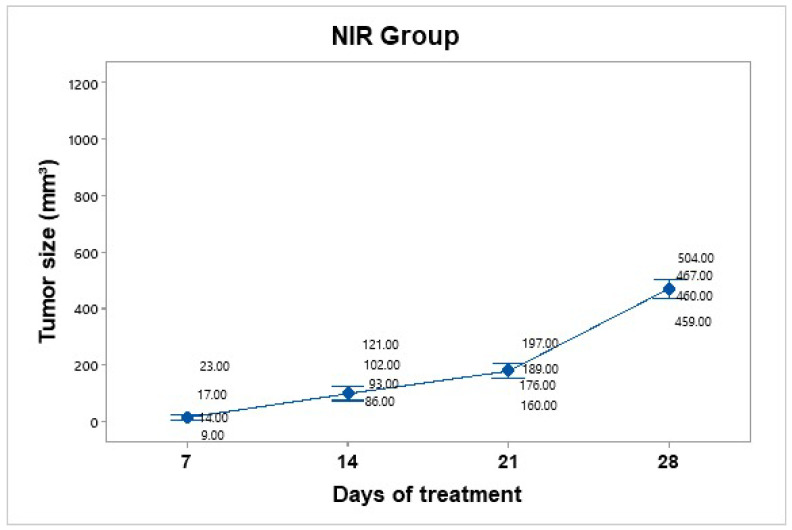
Tumor volume progression in the near-infrared (NIR) irradiation group over 28 days. Although tumor growth was slower than that in the untreated control group, it remained substantial, reaching an average of 472 mm^3^ by day 28. Individual tumor measurements and group means (blue dots) are shown with 95% confidence intervals (Cis) based on Student’s t distribution.

**Figure 9 pharmaceutics-17-00988-f009:**
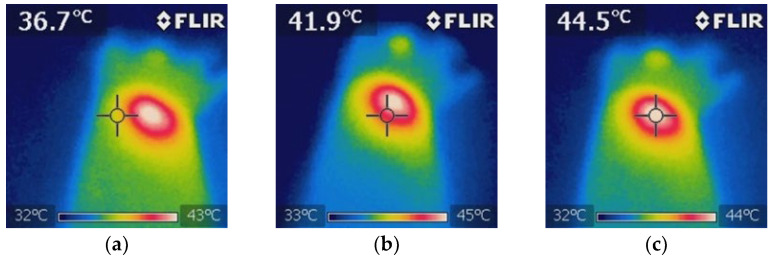
Thermographic analysis of the temperature distribution during NIR irradiation in the BGNS + NIR treatment group. (**a**) The baseline image before irradiation shows a uniform temperature distribution of approximately 36.7 ± 0.5 °C surrounding tissue. (**b**) Mid-irradiation image revealing localized heating at the tumor site, with a visible thermal gradient indicating effective photothermal conversion by BGNSs. (**c**) Final image after 10 min of NIR exposure showing that the peak temperature at the tumor core reached 44.5 ± 0.5 °C, confirming the successful induction of hyperthermia while maintaining the surrounding tissue within safe thermal limits. These images validated the spatial precision and thermal efficacy of the BGNS-mediated PTT.

**Figure 10 pharmaceutics-17-00988-f010:**
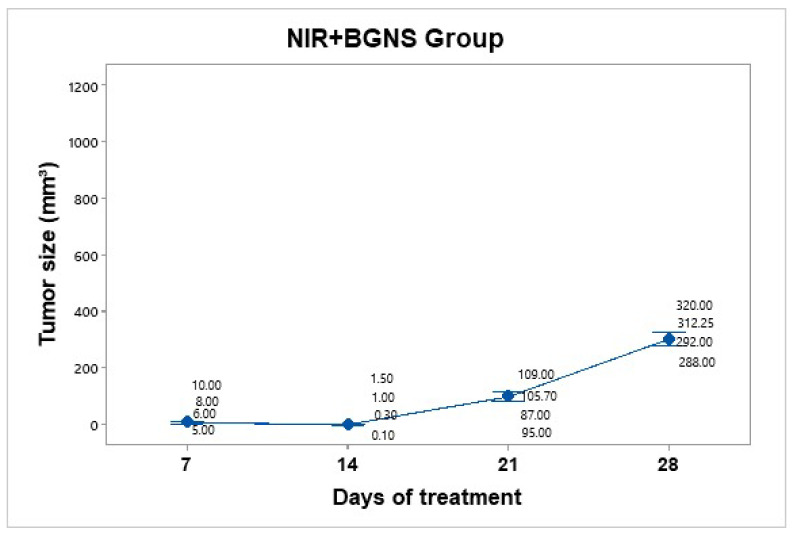
Tumor volume progression in the group treated with BGNS + NIR over 28 days. Although an initial reduction in tumor size was observed, regrowth occurred in subsequent weeks, reaching an average of 303 mm^3^ by day 28. Individual tumor measurementsand group means (blue dots) are shown with 95% confidence intervals (CIs) based on Student’s t distribution.

**Figure 11 pharmaceutics-17-00988-f011:**
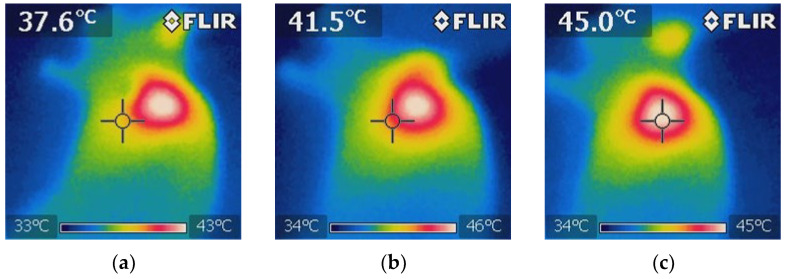
Thermographic images showing the temperature distribution during NIR irradiation in the 5-FU + BGNS + NIR treatment group. (**a**) Initial thermal image before irradiation, showing a uniform temperature of approximately 37.6 ± 0.5 °C surrounding tissue. (**b**) Midpoint of irradiation, where a localized increase in temperature begins to appear at the tumor site, indicating the activation of BGNSs under NIR light. (**c**) Final image after 10 min of irradiation, showing a peak temperature of 45 ± 0.5 °C at the tumor core, confirming effective and targeted photothermal conversion. These images demonstrate the spatial precision and thermal efficiency of the combined chemo-photothermal treatment.

**Figure 12 pharmaceutics-17-00988-f012:**
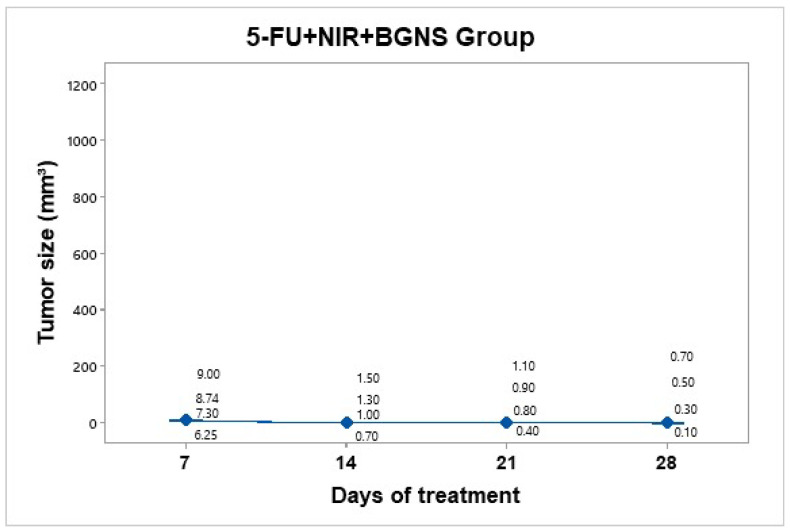
Tumor volume progression in the group treated with combined chemo-photothermal therapy (5-FU + BGNS + NIR) over 28 days. A marked and sustained reduction in tumor size was observed, with an average volume of 0.4 mm^3^ by day 28. Individual tumor measurements and group means (blue dots) are shown with 95% confidence intervals (CIs) based on Student’s t distribution. This group exhibited the most significant therapeutic response among all the treatments.

**Figure 13 pharmaceutics-17-00988-f013:**
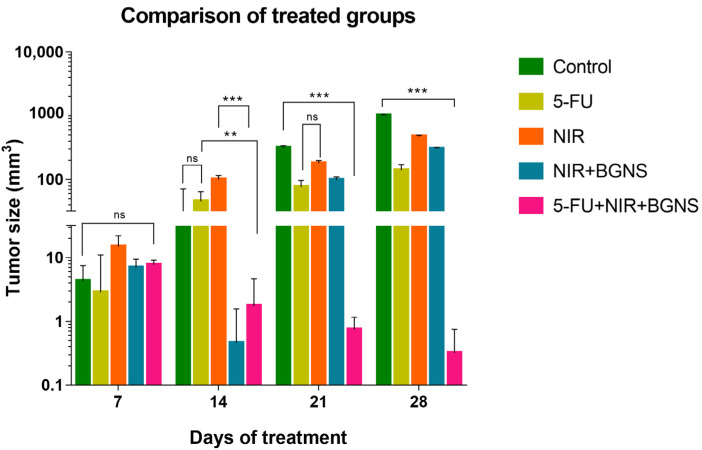
Comparison of tumor volume progression across all treatment groups over 28 days. The control group exhibited exponential tumor growth, reaching an average of 1010 mm^3^. The NIR-only group exhibited moderate growth (472 mm^3^), whereas the BGNS + NIR group initially showed a reduced tumor size but later experienced regrowth (303 mm^3^). The 5-FU group maintained a steady growth curve, with a final average of 142 mm^3^. In contrast, the combined treatment group, consisting of 5-FU, BGNS, and NIR, exhibited the most significant therapeutic effect, with a reduction in tumor volume to 0.4 mm^3^. This figure highlights the superior efficacy of combined chemotherapy and photothermal therapy. Statistical analysis was performed by two-way ANOVA and Tukey’s test (** *p* < 0.05, *** *p* < 0.001). ns: no statistical significance.

**Figure 14 pharmaceutics-17-00988-f014:**
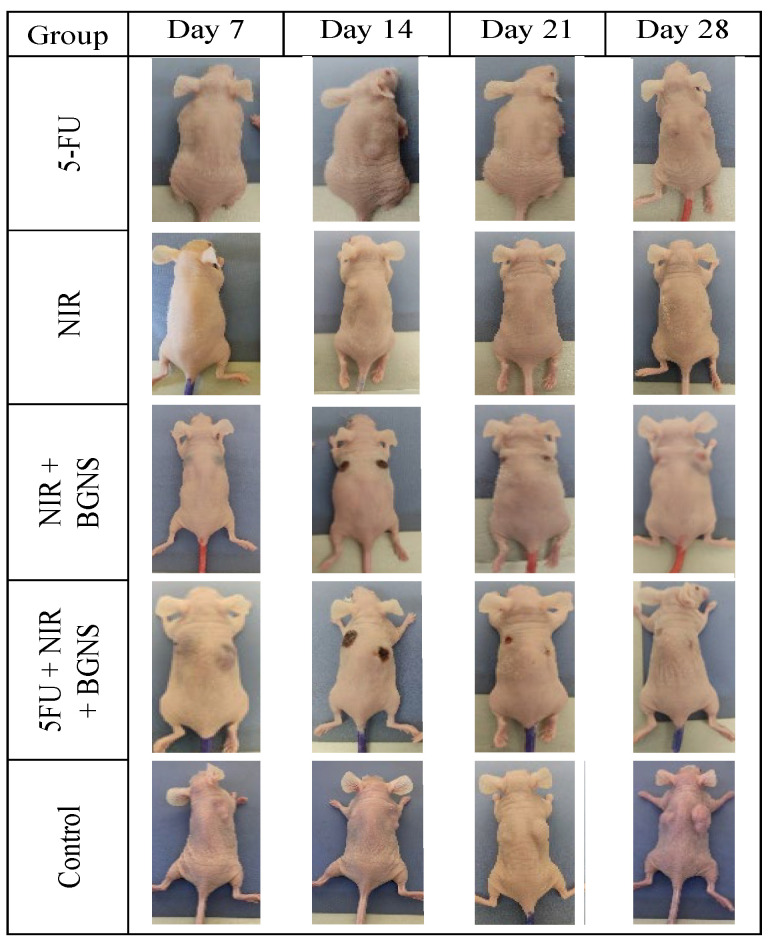
Visual comparison of tumor size between the control group and the complete treatment group (5-FU + BGNS + NIR) at days 7, 14, 21, and 28. The images revealed progressive and uncontrolled tumor growth in the control group, whereas there was marked tumor regression and near-complete disappearance in the combined therapy group. These representative images support quantitative data and highlight the therapeutic efficacy of the chemo-photothermal approach.

**Table 1 pharmaceutics-17-00988-t001:** Hydrodynamic diameter and Z-potential of PLGA precursors and pegylated BGNS. The results are reported as mean ± standard deviation.

Sample	Hydrodynamic Diameter (nm) ^1^	Z-Potential (mV) ^1^
PLGA NPs	126.3 ± 7.1	−12.1 ± 2.6
PLGA-oligochitosan	160.3 ± 2.2	+18.5 ± 1.2
PLGA-seeds	165.9 ± 4.2	+13.4 ± 1.3
BGNS	194.5 ± 2.5	−12.1 ± 0.5

^1^ Mean ± SD.

## Data Availability

The original contributions presented in this study are included in the article. Further inquiries can be directed to the corresponding author(s).

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
