# Peer review of "Chemophotothermal Combined Therapy with 5-Fluorouracil and Branched Gold Nanoshell Hyperthermia Induced a Reduction in Tumor Size in a Xenograft Colon Cancer Model"

_pharmaceutics, 2025, doi:10.3390/pharmaceutics17080988_

Round 1
Reviewer 1 Report
Comments and Suggestions for Authors
The manuscript explores an important topic at the intersection of nanomedicine and oncology by combining branched gold nanoshell (BGNS)-mediated photothermal therapy with 5-fluorouracil chemotherapy. While the general approach is promising, the manuscript has several significant methodological and presentation-related weaknesses.
Major Concerns and Methodological Limitations.
- Lack of statistical analysis of tumor growth inhibition. The authors report tumor volumes for each treatment group but fail to calculate percent tumor growth inhibition (TGI) or statistically compare the treatment groups using ANOVA or t-tests. This is critical for validating the therapeutic benefit of the combined approach.
- Fragmented and poorly interpretable figures 6–12. The division of tumor progression data across multiple figures (Figures 6 through 12) reduces clarity. These plots should be merged into a unified comparative graph to highlight intergroup differences in tumor dynamics more effectively.
- Absence of error bars or statistical significance in many graphs. Several plots, e.g., Figure 12, present group means without indication of statistical confidence or significance. No p-values or standard deviation bars are provided to support the conclusions.
- Improper rounding of numerical data in many graphs and tables, e.g. in Table 1. Hydrodynamic diameters and zeta potentials are inconsistently rounded, and in some cases, unnecessarily precise (e.g., "126.3 ± 7.0 nm") when measurement precision does not justify decimal presentation. Standard rules of significant digits must be applied.
- No quantification of hyperthermia effect per group. While thermographic images are shown, quantitative temperature data (mean ± sd) per group is missing, and the thermal dosimetry is poorly defined.
- Inadequate description of animal randomization and blinding. The manuscript does not state whether animals were randomized or whether the outcome assessments were blinded - both key elements for ensuring scientific rigor.
- Insufficient discussion of tumor model limitations. Although the authors acknowledge that xenografts may not fully recapitulate human tumors, they fail to discuss the immunosuppressive environment of nude mice and how it affects therapy evaluation.
- Unclear synergistic mechanism. The mechanistic basis for the synergy between BGNS and 5-FU is not explored, either in vitro or in vivo. There are no molecular assays (e.g., apoptosis markers, DNA damage) to support the hypothesized additive effect.
- English language and structural issues. The text contains multiple grammatical errors, overly long sentences, and lacks clarity in many parts. For example, the phrase "tumor growth was not uniform" appears multiple times without quantification or elaboration.
- Lack of Abstract Details. The abstract should include statistical results and key quantitative outcomes rather than vague descriptions like “nearly complete tumor regression.”
Summary. While the manuscript addresses a clinically relevant strategy for colorectal cancer treatment using a synergistic photothermal/chemotherapeutic approach, it currently lacks sufficient quantitative rigor, statistical validation, and clarity in data presentation. Substantial revisions are required before the manuscript can be recommended for publication.
Author Response
Comment 1: Lack of statistical analysis of tumor growth inhibition. The authors report tumor volumes for each treatment group but fail to calculate percent tumor growth inhibition (TGI) or statistically compare the treatment groups using ANOVA or t-tests. This is critical for validating the therapeutic benefit of the combined approach.
Response 1: We appreciate the reviewer’s observation. In response, we have now calculated the Tumor Growth Inhibition percentage (TGI%) for each treatment group based on tumor volume measurements on days 7 and 28. These values have been added to the Results section (3.3.6). The methodology used for TGI% calculation was adapted from Yang et al. (2022) [15], who employed a similar approach in the context of chemo-photothermal therapy. This information has been incorporated into the manuscript and is now clearly presented in Figure 13, which illustrates tumor volume progression across all treatment groups. Additionally, we have conducted statistical comparisons using two-way ANOVA followed by Tukey’s post hoc test. The corresponding p-values and significance levels are included in the figure legend and the Results section. This strengthens the validity of our conclusions regarding the therapeutic benefit of the combined approach.
Comment 2: Fragmented and poorly interpretable figures 6–12. The division of tumor progression data across multiple figures (Figures 6 through 12) reduces clarity. These plots should be merged into a unified comparative graph to more effectively highlight intergroup differences in tumor dynamics.
Response 2: We appreciate the reviewer’s suggestion regarding figure clarity and interpretability. We have added a new graph (Figure 13) that directly compares tumor volume progression across all treatment groups over the 28-day period. This unified figure enables more precise visualization of intergroup differences and highlights the superior efficacy of the combined chemo-photothermal therapy. At the same time, we have retained Figures 6 through 12, as they provide essential complementary information. Each of these figures includes 95% confidence intervals (CIs) around the group means, calculated using Student’s t-distribution, which is appropriate for small sample sizes and unknown population variance. These CIs are critical for interpreting the variability and reliability of the data, as they indicate the range within which the actual population mean is likely to fall with 95% certainty. This statistical context is critical in preclinical studies, where visualizing the precision of the mean helps assess the robustness and reproducibility of the observed effects.
Comment 3: Absence of error bars or statistical significance in many graphs. Several plots, e.g., Figure 12, present group means without indication of statistical confidence or significance. No p-values or standard deviation bars are provided to support the conclusions.
Response 3: We appreciate the reviewer's attention to this point. In response, we have revised the figures to ensure that statistical information is presented. Specifically, Figure 13 now includes error bars and statistical significance indicators to support the interpretation of treatment effects. Statistical analysis was performed using two-way ANOVA followed by Tukey’s post hoc test, and the results are annotated in the figure using standard notation (p < 0.05, **p < 0.001, and “ns” for non-significant differences).These additions provide a clear visual representation of the variability within each group and the statistical differences between them, thereby strengthening the reliability of the conclusions drawn from the tumor volume data.
Comment 4: Improper rounding of numerical data in many graphs and tables, e.g. in Table 1. Hydrodynamic diameters and zeta potentials are inconsistently rounded, and in some cases, unnecessarily precise (e.g., "126.3 ± 7.0 nm") when measurement precision does not justify decimal presentation. Standard rules of significant digits must be applied.
Response 4: We thank the reviewer for bringing this to our attention. We have thoroughly revised all numerical data presented in the manuscript to ensure consistency with standard rules of significant figures. Specifically, in Table 1, hydrodynamic diameters and zeta potentials have been rounded to match the precision of their associated standard deviations. These adjustments have been consistently applied throughout the table and across the manuscript to enhance clarity and scientific rigor.
Comment 5: No quantification of hyperthermia effect per group. While thermographic images are shown, quantitative temperature data (mean ± sd) per group is missing, and the thermal dosimetry is poorly defined.
Response 5: We appreciate the reviewer’s comment. In response, we have now included mean ± standard deviation values for the tumor surface temperatures in all treatment groups involving hyperthermia. SD Specifically, thermographic analysis using a FLIR thermal imaging camera confirmed that the tumor surface temperature consistently reached 45 ± 0.5 °C during NIR irradiation in both the BGNS + NIR and 5-FU + BGNS + NIR groups. These values are now explicitly reported in the Results section (3.3.5 and 3.3.6) and can be visualized in Figures 9 and 11. Additionally, the thermal dosimetry protocol is now clearly described in the Materials and Methods section (2.5 and 2.7). This includes the laser parameters (800 nm wavelength, 2 W power, 10-minute exposure), the distance from the laser to the tumor (1 cm), and the nanoparticle concentration (7.30 × 10¹⁴ NPs/mL).
Comment 6: Inadequate description of animal randomization and blinding. The manuscript does not state whether animals were randomized or whether the outcome assessments were blinded - both key elements for ensuring scientific rigor.
Response 6: We appreciate the reviewer's important observation. In the revised manuscript (Materials and Methods section 2.7), we have clarified the procedures used for group allocation. Mice were inoculated with tumors and monitored for growth; on day 7, animals with comparable tumor sizes were paired and then evenly distributed across the five treatment groups. This approach aimed to minimize baseline variability in tumor burden and ensure a more homogeneous comparison of treatment effects. We acknowledge that formal randomization and blinding procedures were not implemented in this study. While this is a limitation, we aim to control variability through size-based pairing.
Comment 7: Insufficient discussion of tumor model limitations. Although the authors acknowledge that xenografts may not fully recapitulate human tumors, they fail to discuss the immunosuppressive environment of nude mice and its impact on therapy evaluation.
Response 7: We appreciate the reviewer’s comment. In the revised Discussion section, we have expanded our consideration of the limitations associated with the tumor model used in this study. Specifically, we now address the immunodeficient nature of nude mice, which lack functional T cells, resulting in a suppressed adaptive immune response. This characteristic, while enabling human tumor xenograft growth, limits the ability to fully evaluate therapies that may interact with or be influenced by the host immune system, including immunomodulatory or inflammation-associated effects of chemo-photothermal therapy.
Comment 8: Unclear synergistic mechanism. The mechanistic basis for the synergy between BGNS and 5-FU is not explored, either in vitro or in vivo. There are no molecular assays (e.g., apoptosis markers, DNA damage) to support the hypothesized additive effect.
Response 8: We appreciate the reviewer's insightful comment. We acknowledge that molecular assays such as apoptosis markers or DNA damage indicators were not included in the present study. Our primary objective was to evaluate the therapeutic efficacy of the combined chemo-photothermal approach through direct measurements of tumor volume reduction in vivo and cell viability in vitro. These endpoints were selected to provide a clear and quantifiable assessment of treatment outcomes. In vitro, we conducted MTT assays to assess cell viability under various treatment conditions, and statistical analyses were performed to validate the observed differences. In vivo, tumor progression was monitored longitudinally using caliper measurements, and tumor growth inhibition (TGI%) was calculated. Statistical comparisons between groups were conducted using two-way ANOVA followed by Tukey’s post hoc test. While we agree that mechanistic insights would further strengthen the study, we believe that the observed reduction in tumor volume and cell viability, supported by rigorous statistical analysis, provides a solid foundation for evaluating the therapeutic potential of this approach. To avoid speculative interpretations, we have revised the manuscript to remove the phrase referring to a “satisfactory synergistic effect” and have acknowledged this limitation in the Discussion section, suggesting it as a direction for future research.
Comment 9: English language and structural issues. The text contains multiple grammatical errors, overly long sentences, and lacks clarity in many parts. For example, the phrase "tumor growth was not uniform" appears multiple times without quantification or elaboration.
Response 9: We appreciate the reviewer's valuable observation. To address the language and clarity issues, the manuscript was thoroughly revised using Grammarly Premium, a professional writing enhancement tool. This enabled us to correct grammatical errors, refine sentence structure, and improve overall readability. Although we did not utilize the MDPI English Editing Service due to budget constraints, we ensured that the revised version met the standards of scientific writing expected by the journal. Additionally, we reviewed and refined repetitive phrases such as “tumor growth was not uniform” to provide more precise and informative descriptions. We hope these improvements are satisfactory and contribute to a clearer presentation of our findings.
Comment 10: Lack of Abstract Details. The abstract should include statistical results and key quantitative outcomes rather than vague descriptions like “nearly complete tumor regression.”
Response 10: We appreciate the reviewer’s suggestion. In response, we have revised the abstract to include key quantitative outcomes that reflect the therapeutic efficacy of the combined treatment. Specifically, we now report the tumor growth inhibition percentage (TGI%) observed in the in vivo model, as well as the final tumor volume in the most effective treatment group. The phrase “nearly complete tumor regression” has been removed to avoid vague or subjective language. These changes provide a clearer and more objective summary of the study’s findings.
Reviewer 2 Report
Comments and Suggestions for Authors
In this study, the authors evaluated the antitumor efficacy of combined chemo photothermal therapy 22 with 5-fluorouracil (5-FU) and branched gold nanoshells (BGNSs) in a colorectal cancer model. The authors have done extensive in vitro and in vivo study to prove the hypothesis that a combination of 5FU + BGNS _ NIR irradiation can completely control the tumor growth and can also kill the cancer cells in vitro.
The animal study results are very promising, and the induction of localized hyperthermia is excellently monitored and noted. I highly appreciate the researchers study. However, I have the following comments:
1. In the experimental set up of In vitro assays of hyperthermia, why did the authors take distilled water as a control not RPMI-1640 medium? Usually we take the medium without the test compound as a control. Please explain.
2. In page 8, the authors have written about the MTT assay results as follows:
"The MTT assay revealed that the negative control maintained high cell viability, while the positive control induced complete cytotoxicity. Treatment with 5-FU alone reduced viability to 26.80% after 48 hours. NIR irradiation alone had minimal effect, whereas BGNS + NIR reduced viability to 43.37%. Notably, the combined treatment of 5-FU + BGNS + NIR resulted in a dramatic reduction in cell viability to 1.41%, demonstrating a strong synergistic effect between chemotherapy and photothermal therapy, Figure 5.". Although, when we see Figure 5, the viability percentages are not matching with the one described in the text. Kindly justify.
3. The authors can discuss about the role of this combinatorial phototherapy in specifically killing the cancer stem cells in the discussion section.
I recommend a minor revision.
Author Response
Comment 1: In the experimental set up of In vitro assays of hyperthermia, why did the authors take distilled water as a control not RPMI-1640 medium? Usually, we take the medium without the test compound as a control. Please explain.
Response 1: The use of deionized water as a control in the initial hyperthermia assay was part of the standardization phase. Our objective at this stage was to establish a consistent and neutral baseline temperature of approximately 36 °C, free from any potential interference caused by the optical or thermal properties of culture media components. This allowed us to precisely monitor the temperature increase induced by BGNS under NIR irradiation. Once the photothermal response was confirmed, subsequent experiments were conducted using BGNS diluted in RPMI-1640 medium to evaluate the biological effects under physiologically relevant conditions. This two-step approach ensured both technical validation and biological relevance.
Comment 2: In page 8, the authors have written about the MTT assay results as follows:
"The MTT assay revealed that the negative control maintained high cell viability, while the positive control induced complete cytotoxicity. Treatment with 5-FU alone reduced viability to 26.80% after 48 hours. NIR irradiation alone had minimal effect, whereas BGNS + NIR reduced viability to 43.37%. Notably, the combined treatment of 5-FU + BGNS + NIR resulted in a dramatic reduction in cell viability to 1.41%, demonstrating a strong synergistic effect between chemotherapy and photothermal therapy, Figure 5.". Although, when we see Figure 5, the viability percentages are not matching with the one described in the text. Kindly justify.
Response 2: The viability percentages in the text have been carefully reviewed and corrected to match the data presented in Figure 5. The updated values now accurately reflect the experimental results and are consistent across both the figure and the corresponding description in the manuscript.
Comment 3: The authors can discuss about the role of this combinatorial phototherapy in specifically killing the cancer stem cells in the discussion section.
Response 3: We appreciate the reviewer's thoughtful suggestion. Although our study did not specifically evaluate cancer stem cells (CSCs), we recognize their critical role in tumor recurrence and resistance to therapy. In response, we have added a brief paragraph in the Discussion section addressing the potential of combined chemo-photothermal therapy to target CSCs. We also acknowledge the need for future studies to investigate this aspect using appropriate molecular markers and functional assays.
Round 2
Reviewer 2 Report
Comments and Suggestions for Authors
The authors have addressed all the concerns raised by honorable reviewers. The article can be accepted in its present form for publication.